# Bullous Central Serous Chorioretinopathy: A Rare and Atypical Form of Central Serous Chorioretinopathy. A Systematic Review

**DOI:** 10.3390/ph13090221

**Published:** 2020-08-28

**Authors:** Francesco Sartini, Martina Menchini, Chiara Posarelli, Giamberto Casini, Michele Figus

**Affiliations:** Ophthalmology, Department of Surgical, Medical, Molecular Pathology and of the Critical Area, University of Pisa, 56126 Pisa, Italy; martina.mmenchini@gmail.com (M.M.); chiaraposarelli@gmail.com (C.P.); giamberto.c@alice.it (G.C.); figusmichele@gmail.com (M.F.)

**Keywords:** bullous central serous chorioretinopathy, chronic central serous chorioretinopathy, exudative retinal detachment, argon laser photocoagulation, mineralocorticoid receptor antagonists, transpupillary thermal therapy, pachychoroid, scleral thinning surgery

## Abstract

Bullous central serous chorioretinopathy (bCSCR) is a rare variant of the central serous chorioretinopathy, complicated by an exudative retinal detachment with shifting fluid. This systematic review aims to present the epidemiology, the pathogenesis, the clinical presentation, the imaging, the differential diagnosis, and the latest treatments of this disease. A total of 60 studies were identified following a literature search adhering to PRISMA guidelines. After full-text evaluation, 34 studies about bCSCR were included. bCSCR usually affects middle-aged men, and the principal risk factor is corticosteroid medications. Pathogenesis is related to an increased choroidal vessel and choriocapillaris permeability, with subsequent subretinal fluid accumulation, rich in fibrin, which may provoke the exudative retinal detachment. Clinical presentation and imaging are fundamental to distinguish bCSCR from other pathologies, avoiding unappropriated treatment. Corticosteroid withdraws (if assumed) and laser photocoagulation of leakage sites seen at angiography may speed up retinal reattachment. Verteporfin photodynamic therapy, transpupillary thermal therapy, oral eplerenone and scleral thinning surgery are other therapeutic options. An early diagnosis might prevent disease progression due to harmful medications as well as unnecessary surgery.

## 1. Introduction

Bullous central serous chorioretinopathy (bCSCR) is a rare and atypical form of chronic central serous chorioretinopathy (CSCR) [1]. It was firstly described by Gass in 1973 [2]. Patients affected usually had multiple pigment epithelial detachment (PED), hidden beneath a cloudy subretinal fluid (SRF) [3]. When a PED evolves into retinal pigment epithelium (RPE) tear, an exudative retinal detachment (ERD) may develop, becoming a clinical challenge [3]. Thus, bCSCR can be misdiagnosed as a rhegmatogenous retinal detachment (RRD) or an ERD, secondary to other pathologies such as Harada’s disease [1]. 

To date, bCSCR epidemiology, aetiopathogenesis and optimum treatment are still unclear.

This systematic review aims to analyze data available on bCSCR, focusing on the epidemiology, pathogenesis, clinical presentation, imaging, differential diagnosis and currently available treatments. 

## 2. Results

### 2.1. Epidemiology

Age of bCSCR onset ranges from 27 to 63 years (mean 43.1 years), with a predilection for male, as classic CSCR [4]. Bilateral involvement is more frequent in bCSCR compared to chronic CSCR (84% respectively 7.4%) [4,5]. Hispanic and Asian ethnicities seem to have an increased risk to develop bCSCR compared to other ethnicities, but further studies are warranted [5]. In the vast majority of patients, it occurs since the beginning; however, about one-third of cases have a previous history of CSCR (7 months to 9 years before) [4]. bCSCR can recur in about half of cases if not treated [4]. To sum up, no significant differences were reported between bCSCR and chronic CSCR in terms of age, sex and ethnicity [6]. 

### 2.2. Pathophysiology

Various risk factors have been reported in bCSCR pathophysiology: systemic corticosteroid therapy (e.g., for Chron’s disease, ulcerative colitis, severe asthma), organ transplantation, haemodialysis, pituitary adenoma and pregnancy [3,5,7,8,9,10,11,12]. Interestingly, also epidural steroid injection for chronic back pain is reported as bCSCR risk factor [13]. 

Choroidal vascular dysfunction is a crucial feature in the bCSCR pathophysiology, as in CSCR [1]. In particular, marked congestion and breakdown in the choroidal vessel’s permeability alter RPE, damaging RPE outer blood-retina barrier [1]. Subsequently, proteins and fibrinogen leaks, thus a PED and/or a subretinal exudation occur [1]. Both chronic CSCR and bCSCR shows pachychoroid feature as outer choroidal vessel dilation, inner choroid atrophy, and choroidal hyperpermeability; however, to date, the mechanism responsible for bCSCR development is still unclear [1,14]. It has been speculated that sub-EPR fibrin and increased hydrostatic pressure internal to PED provoke circumferential traction on the RPE layer and a subsequent RPE tear [6]. 

Consequently, the intense subretinal fluid accumulation, rich of fibrin and weighty, provokes an ERD, whose fluid shifts according to the patient’s position [4]. Moreover, steroid (systemic or epidural) may play a role in impending the healing of RPE tear and increasing choriocapillaris’ permeability [13]. Nevertheless, no difference in corticosteroid exposure was reported between bCSCR and chronic CSCR patients [6].

Interestingly Kowalczuk et al. compared proteins and metabolites of the SRF obtained from patients affected by bCSCR or RRD [15]. Downregulation of the alternative complement pathway was found in bCSCR. It is involved in physiologic transport of ions and macromolecules, and it might cause fluid accumulation into subretinal space [15]. Furthermore, induced gluconeogenesis and glycolysis in bCSCR may promote cone photoreceptor’s survival [15]. Also, the farnesoid X receptor (FXR) pathway activated in bCSCR, highlighted other neuroprotective mechanisms. Additionally, proteins involved in cell migration and adhesion (e.g., opticin, myocilin, metalloproteinase-2), usually abundant in the vitreous, were reduced in bCSCR, justifying the better architecture preservation in this disease [15]. Finally, downregulation of steroid metabolism and microglia and lipid metabolism different activation were reported, confirming the different aetiopathogenesis between bCSCR and RRD [15]. 

### 2.3. Clinical Presentation

The patient usually experiences metamorphopsia and central scotomata [11]. Best-corrected visual acuity (BCVA) is reduced, ranging from 20/25 to 20/200 at presentation [5].

The anterior segment usually has no inflammation signs, and the vitreous is clear [3]. 

Fundus examination reveals a yellow-white subretinal exudation at the posterior pole associated with an ERD, usually inferior, without retinal hemorrhages, acute chorioretinitis, retinal tear or choroidal detachment [3,4,11]. The typical lesion appears doughnut-shaped with a dark brown central area surrounded by yellowish cloudy exudates [4]. Close or underneath the SRF, one or more PED can be observed, usually more extensive and more numerous than ones seen in chronic CSCR [3]. Nevertheless, the underlying RPE layer often is hidden by the cloudy SRF [6]. Notably, ERD shifts to the posterior pole when the patient assumes a supine position [4]. RPE tear, usually seen in bCSCR patients (95% of the affected eyes vs. 9% in classical CSCR), is located at the site of a previous PED, and appears as a dark-grey colored lesion, with crescent or polygonal shape without retracted edges, named concentric type [6] (Figure 1).

As in chronic retinal detachment, subretinal fluid can cause retinal ischemia, reducing oxygenation of outer retinal layers [16]. Therefore, peripheral non-perfusion can occur, and telangiectatic changes arising from terminal capillaries are observable [17,18]. Moreover, the elevated concentration of fibrin in SRF predisposes to the development of subretinal fibrosis, which may impair the visual outcome [19]. 

Comparing chronic CSCR and bCSCR, best-corrected visual acuity at the baseline and the final visit shows no statistical difference in a retrospective review of patients [6]. Although RPE tear and peripheral nonperfusion occur only bCSCR and also PEDs are significantly more frequent [6]. Moreover, on cursory examination, the ERD can simulate diseases not belonging to the CSCR spectrum, leading to inappropriate therapeutic procedures [14].

Recurrence of bCSCR is frequent, but finally, it becomes quiescent; thus, multiple atrophic scars associated or not peripheral retinal atrophic tract are detectable [4]. The visual outcome is severely impaired if the macula is involved [4].

### 2.4. Imaging

At the level of subretinal exudation, optical coherence tomography (OCT) detects a dome-like detachment of thickened neurosensory retina associated with a semi-transparent subretinal space, due to the high concentration of fibrin in the subretinal fluid [4]. PED usually have an internal hyperreflectivity, but also internal hyporeflectivity are detectable [6]. A disruption to the external limiting membrane and ellipsoid zone, retinal folds or subretinal fibrin may be frequently detected [6]. OCT also shows RPE tear as a dense hyperreflectivity zone, corresponding to RPE duplication, next to a less intense hyperreflective area due to the bare choroid [20] (Figure 2).

Finally, with enhanced depth imaging (EDI) OCT, hyperreflectivity around large choroidal vessels and at the choriocapillaris are detected more frequently than in classical CSCR [6]. Although, there is no significative difference about subfoveal choroidal thickness between classical CSCR and bCSCR [6].

Near-infrared images disclose irregular, notable hyper-reflectivity at the scrolled edge of the RPE tear, and homogeneous, slight hyper-reflectivity in the denuded zone [20].

Fundus autofluorescence reveals an hyperautofluorescence due to the RPE tear initially with sharp border, which became blurry in time [21]. Besides, confluent hypoautofluorescence, granular hypoautofluorescence, granular hyperautofluorescence and descending tracts are observable [6] (Figure 3).

In the phases preceding bCSCR development, fluorescein angiography (FA) reveals focal areas of hyperfluorescence (blowouts) within PEDs [6].

When bCSCR develops, FA highlights multiple RPEs tears as early hyperfluorescence and late staining (2–22, mean 6.7), at sites correlating to the position of previous PEDs [6]. The retina is attached to the area devoid of RPE by subretinal fibrin at sites of RPE tears [6]. A pooling of the sub-neurosensory retinal space is also highlighted [3]. This hyperfluorescence descends inferiorly conforming to the dependent retinal detachment [4]. Furthermore, slow and rather diffuse bilateral leakage attest diffuse decompensation of the RPE [22]. In particular, the crescent RPE tear appears as a window defect (the bare choroid) next to an hypofluorescence (the rolled RPE layer), in all angiographic phases if it has a crescent shape type [20]. Conversely, concentric RPE tear appears as an island of retracted RPE tissue at the center of the tear, with a hyperfluorescent area all around [6]. 

In the case of peripheral retinal ischemia and neovascularization, wide-field FA can detect retinal capillaries obliteration and neovascularization with profuse leakages at the junction between perfused and non-perfused retina during the late angiographic phase [18,22]. Finally, no disk leak or vasculitis are detected [14]. 

Indocyanine green angiography (ICGA) shows dilated choroidal vessels with choroidal hyperpermeability, and in 20% of cases, a filling delay of choroidal circulation [23,24]. In late phases, PED demonstrates persistent hypofluorescence [13] (Figure 4).

Finally, OCT angiography (OCTA) can detect choroidal neovascularization in chronic CSCR; however, its role in bCSCR is still being explored [1]. In particular, OCTA shows ill-defined low detectable flow areas (dark areas) corresponding to SRF and well-delineated areas with no detectable flow (dark spots) at the choriocapillaris level, corresponding to PED [1]. Furthermore, OCTA depicts the absence of any vascular network below the RPE tear [14]. Finally, OCTA highlights the pachychoroid pattern in bCSCR, as well-delineated, high-flow, tangled pattern areas within the choriocapillaris overlying dilated outer choroidal vessels [14].

### 2.5. Differential Diagnosis

Distinguishing bCSCR from RRD or ERD can be challenging [14,25]. 

RRD can be excluded if the following symptoms and signs are present: central vision impairment occurring before peripheral scotoma, lack of photopsia or floaters, absence of retinal breaks, lack of tobacco dust into the vitreous, shift of subretinal fluid, angiographic evidence of PED under neurosensory retinal detachment [26].

Leading causes of ERD are Harada’s disease, posterior scleritis, choroidal tumor, or uveal effusion syndrome, to avoid uncorrected treatments as vitrectomy or corticosteroid therapy [14,25].

Harada’s disease has no sex predilection, and the patient may experience nausea, vomiting, headaches or malaise, preceding bilateral RD. Typical signs include vitreous inflammation, mild iridocyclitis, and it responds well to corticosteroids [25,26]. 

Posterior scleritis occurs more frequently in women, and 50% of cases are bilateral. Ocular pain often accompanies the disease, which is caused by fungi, bacteria, viruses, relapsing polychondritis, rheumatoid arthritis and systemic lupus erythematosus [26].

Choroidal tumor and metastasis can be easily excluded with fundus examination, but in challenging cases, imaging, like ocular ultrasound, can be helpful [14,26].

Finally, uveal effusion syndrome affects prevalently male, as bCSCR, although optic disc oedema and a flat choroidal detachment in the fundus periphery can be observed [26].

### 2.6. Treatment

Several studies reported a self-resolving course of bCSCR, although recurrent [3,4,27]. Conversely, bCSCR course longer than a year if non treated has been reported, with development of subretinal proliferation, RPE degeneration and atrophy [23]. However, final BCVA widely ranges from better than 20/40 to worse than 20/200 [3,4,27,28].

As in CSCR, the treatment aims to preserve the outer neurosensory retinal layers, achieving a lasting and complete SRF resolution, because even a small amount of SRF can damage the photoreceptors [29]. Elimination of risk factors is advisable in bCSCR as in CSCR. In particular, systemic corticosteroid withdrawal is suggested as the first step, if possible, according to systemic conditions (e.g., previous organ transplant) [9,18,25,28,30,31,32]. Kunavisarut et al. stopped steroid medication in six patients and tapered it in one (adding azathioprine). Retinal reattachment occurred in 5 months (range 1–9 months) in 86% of cases [30].

The treatment of chronic CSCR is still debated, but recently evidence-based guidelines have been proposed [29]. After individuation and elimination of risk factors, if persistent SRF and one or more focal leakage points on FA are present, ICGA- and FA-guided half-dose (or half fluence) verteporfin photodynamic therapy (PDT) is suggested as first-line treatment. Then, if SRF persists FA, ICGA and OCTA should be obtained. If choroidal neovascularization is detected, intravitreal injections should be performed, with or without adjunctive PDT. If leakage (diffuse or focal) is noted on FA/ICGA, the following treatment can be performed: retreatment with PDT, subthreshold micropulse laser, mineralocorticoid receptors antagonist, observation. Finally, if ICGA hyperfluorescence or leakage on FA are not detectable, retreatment is not advisable, because no concrete evidence is available [29].

Conversely, the bCSCR therapeutic algorithm is still a matter of controversy, due to its rare frequency and the lack of randomized clinical trial to compare different treatment results. After risk factor removal, laser photocoagulation on focal leakage seen on FA can be performed [8,23,32]. Nevertheless, if the leakage area is extended, laser photocoagulation can lead to scotomata. Therefore other therapeutic approaches have been described, such as PDT, oral eplerenone, and transpupillary thermal therapy (TTT) [33]. Finally, anti-vascular endothelial growth factor (VEGF) intravitreal injections and surgery have been reported in the literature [14,21].

#### 2.6.1. Argon Laser Photocoagulation

Argon laser photocoagulation to the focal leaks seen at FA may accelerate bCSCR resolution [8,23,32]. Sahu et al. compared retrospectively this treatment versus observation in 11 patients affected by bCSCR [3]. In the first group, bCSCR resolution occurred in 12 weeks with a final BCVA better than 20/30 in 75% of cases, instead of observation group obtained clinical resolution in 14 weeks and a BCVA better than 20/30 in 89% of cases. Thus the same authors concluded that laser treatment did not offer additional benefit over the natural course of the disease [3]. Otsuka et al. enrolled 25 patients affected by bCSCR over a mean follow-up of 10.6 years [4]. In particular, 8 patients showed a spontaneous clinical resolution in a few months (not specified). Otherwise, 17 patients underwent laser photocoagulation to the focal leaks (8 single treatment and 9 repeated treatments) with resolution within two months. However, the final BCVA was comparable between the treated and untreated group. Cumulative results about BCVA were 20/20 or better in 24 eyes (52%) and 20/200 or less in 4 eyes (8.7%), due to macular atrophy [4]. Sharma and Colleagues treated 29 bCSCR patients with laser photocoagulation, obtaining a retinal reattachment in all eyes and a BCVA improvement >2 lines in 68.9% of patients [28]. To date, no large, prospective randomized controlled trials have been conducted in order to compare the efficacy or long-term outcome between argon laser photocoagulation versus observation in bCSCR treatment.

#### 2.6.2. PDT

Wykoff et al. reported a case of bCSCR treated with reduced-fluence PDT, who had already received one intravitreal injection of Bevacizumab (1.25 mg) and one of Ranibizumab (0.5 mg) with no clinical improvement [21]. PDT settings were 6mg/m^2^ verteporfin with a 6.3 mm laser spot-size for 83 s at 95 mW and treatment encompassed the macula and all the hypercianescent areas detected at the ICGA [21]. A complete resolution of the retinal detachment was obtained, although BCVA decreased from 20/60 to 20/80 seven months after the treatment, due to the disruption of the photoreceptor inner segment/outer segment reflective band within the fovea. A pigmented retinal scar was also developed at the site of the RPE tear [21]. Besides, Ng et al. treated a patient with half-dose PDT (3 mg/m^2^), who already had assumed a two-week course of oral acetazolamide (250 mg QID) [24]. The spot was set to 4.5 mm, covering the macula and dilated choroidal vessels at ICGA. BCVA improved from 20/63 to 20/25 and OCT confirmed the resolution of ERD after three months. At the end of follow-up (38 months), BCVA was 20/20, and neurosensory retinal detachment did not recur [24]. Additional prospective trials are necessary to determine which PDT modality is better for bCSCR treatment [14].

#### 2.6.3. Mineralocorticoid Receptor Antagonist

Mineralocorticoid receptor antagonists may reduce choroidal vasodilation and SRF accumulation, downregulating the vasodilator potassium channel KCa2.3 [34]. 

Oral eplerenone treatment of bCSCR has been reported by Aggarwal et al. [35]. A patient with prolonged use of betamethasone ointment for skin allergy was treated with 25 mg/day eplerenone per os. A BCVA improvement and SRF resolution were obtained at 12 weeks of follow-up [35].

Instead, Ramos-Yau et colleagues treated a 45-year-old female with 50 mg spironolactone per day [36]. Initially managed as a multifocal choroiditis with oral and peribulbar corticoids, the retinal detachment increased and BCVA dropped fifteen days later. Thus, a presumptive diagnosis of bCSCR was made, and treatment with spironolactone started, stopping corticosteroids. After two months, SRF resolved, and BCVA improved from 20/70 to 20/40 (20/20 at presentation). Two months later, oral spironolactone dosage shifted to 25 mg/day, although the further period treatment was not specified [36].

Cebeci et al. combined PDT and laser photocoagulation with 25 mg oral eplerenone twice daily [31]. In particular, a patient affected by bilateral bCSCR was treated with reduced-fluence PDT (25 J/cm^2^, 3000 mW/cm^2^) in the right eye (RE) seen at ICGA and with laser photocoagulation left eye (LE). The ERD was reduced at one month in the RE and resolved in the LE at one month of follow-up. BCVA improved from light perception to counting finger in RE and from 20/32 to 20/25 in the LE [31]. Finally, further studies are warranted to define the efficacy and duration of this treatment.

#### 2.6.4. TTT

TTT delivers heats to pigmented cells of RPE and choroid with low power for a prolonged period, sparing the sensory retina [34]. Kawamura et al. treated eight eyes of eight patients with TTT [33]. In particular, the spot size was determined by FA, and the power setting was set 10% lower than the protocol for neovascularization. Initial average BCVA was 20/80 and at the end of follow-up (4.5 months) was 20/32. Resolution of serous retinal detachment was obtained in 62.5% of cases. In the other patients, retreatment with the same settings was performed, obtaining resolution only in one of three cases. No severe adverse events were reported, just an extrafoveal scar close to the irradiation area [33]. Additional prospective trials will help determine TTT feasibility as a treatment modality for bCSCR.

#### 2.6.5. Surgical Techniques

Surgical options can be divided into two categories: external (transscleral) drainage and internal drainage. 

Regarding transscleral techniques, Venkatesh et al. performed two partial-thickness (4 × 3 mm) scleral resection, in the inferior quadrants, 4mm posterior to lateral recti insertions [37]. Then, mitomycin C 0.02% was placed on the scleral bed for 2 min, followed by a copious wash with balanced saline solution. BCVA improved from 20/630 to 20/200, and the ERD resolved four months after the procedure, with no recurrence in the following two years [37]. Maggio et al. proposed a similar technique (scleral thinning surgery) to reach a rapid and long-lasting resolution of bCSCR [14]. Once exposed the sclera with an inferior conjunctival peritomy, two 4 × 4 mm almost full sclerotomies were done in the inferior quadrants, avoiding the areas of vortex veins exits. Finally, the conjunctiva was re-approximated. They treated a 65-year-old man with a BCVA of 20/63. He obtained a complete SRF resolution the day after surgery and a BCVA of 20/80 with no recurrence at six-months of follow-up [14]. Even if the external drainage techniques avoid the risks related to intraocular surgery, carry the threat of subretinal or choroidal hemorrhages [38].

About internal drainage techniques (pars plana vitrectomy, scleral buckling with cryopexy and retinotomy) have been successfully employed in RD associated to uveitis and may ensure to remove all subretinal fluid [27,38,39,40]. Nevertheless, these treatments often have been reported as a result of misdiagnosed bCSCR [14]. 

John et al. performed an encircling scleral buckle, phacoemulsification and 20-gauge pars plana vitrectomy in a patient affected by bCSCR with BCVA of 20/80 [38]. SRF was drained trough a superior retinotomy, then endolaser on 360° and silicone oil tamponade were applied. One year later, the silicone oil was removed, OCT showed no SRF but minimal intraretinal fluid and BCVA was 20/200 [38]. 

Adan and Corcostegui combined pars plana vitrectomy and injection of perfluorocarbon liquid with trans-scleral drainage and endolaser photocoagulation to leakage points in a patient with bilateral bCSCR [39]. At one year of follow-up, the retina was attached in both eyes, and BCVA was 20/80 in the RE and 20/200 in the LE (initial BCVA not specified) [39]. The same surgical technique was used by Chen at al. in a patient with bCSCR, obtaining retinal reattachment, although BCVA did not improve after six months (counting fingers) [27]. Pars plana vitrectomy combined with transscleral drainage does not require a retinotomy, allowing an immediate retinal reattachment and laser photocoagulation with an improved view through perfluorocarbon liquid; however, reported BCVA is poor [27,39]. 

Finally, Kang et al. treated a patient with lensectomy, pars plana vitrectomy, drainage retinotomy, fluid-gas exchange and endolaser photocoagulation at leakage points [40]. Post-operatively the retina was attached, but the BCVA did not improve. These authors speculated that visual outcome was limited due to long-lasting neuroretina detachment and suggested to perform surgical treatment before subretinal proliferation developed [40]. Therefore, prospective randomized controlled trials are warranted in order to evaluate further the efficacy and safety of surgical techniques to treat bCSCR.

#### 2.6.6. Anti-VEGF Intravitreal Injections

Inhibiting VEGF has an anti-proliferative and anti-hyperpermeability effect on choroidal endothelial cells; however, anti-VEGF intravitreal injections is off-label in CSCR [29]. To date, only in one case report has bCSCR been treated with intravitreal injection of anti-VEGF, with no clinical improvement [21]. In particular, the patient received one injection of Bevacizumab (1.25 mg) followed by one of Ranibizumab (0.5 mg) without performing a loading dose [21]. Besides, peripheral neovascularization complicating bCSCR, have been successfully treated with argon laser photocoagulation of leakage sites [22]. Therefore, further studies are warranted to clarify the possible anti-VEGF therapeutic role in bCSCR.

## 3. Discussion

bCSCR is a rare and atypical form of CSCR, usually affecting middle-aged men otherwise healthy. Systemic corticosteroid assumption is the principal risk factor reported, although bCSCR seems to be the result of environmental, genetic and biomedical interactions. Moreover, pathogenesis is unclear; choroidal vessels hyperpermeability allows fluid accumulation into choroid with subsequent decompensation of RPE. Thus, PED and leakage in subretinal space plentiful of fibrin occur, causing a retinal tear and an ERD. 

Diagnosis of bCSCR can be a dilemma, but it is crucial to distinguish it from other pathologies, in particular Harada’s disease, to avoid unappropriated corticosteroid treatment. However, absence of vitreous cells, shifting nature of the subretinal fluid with fluctuating visual acuity, absence of retinal tears and confinement of the yellow-white lesion to the outer retinal layers with single or multiple PED suggest a bCSCR diagnosis.

OCT detects easily the subretinal fluid with fibrin, hyperreflective PED and the RPE tears. Furthermore, FA and ICGA reveal multiple hyperfluorescence focal leaks and the RPE tear, associated with hyperpermeable and dilated choroidal vessels.

Treatment is based on corticosteroid withdrawal and argon laser photocoagulation to the leakage sites seen on FA. However, if argon laser is not suitable, PDT or TTT can be an option. As of the last choice, scleral thinning surgery may be considered, because it is less invasive with a reduced rate of complication, compared to other surgical techniques. Anti-VEGF intravitreal injections have shown no clinical benefit in the only case reported in the literature. Nevertheless, a long-term follow-up of these patients is mandatory due to the chronicity of bCSCR.

## 4. Materials and Methods

A comprehensive search of the PubMed database was performed on 1 June 2020. Keywords used for the search was “bullous central serous chorioretinopathy”, “bullous CSC” and “bullous CSCR”. The search workflow was designated in adherence to the preferred reporting items for systematic reviews and meta-analyses (PRISMA) statement [41] (Figure 5). The process applied for this review consisted of a systematic search of all available articles regarding bCSCR.

After preparation of the list of all electronic data captured, two reviewers (F.S. and M.M.) examined the titles and abstracts independently and identified relevant articles. Moreover, the reference lists of identified articles were checked manually to detect any potential studies. All studies available in the literature reporting original data on bCSCR were initially included without restriction for study design, sample size and intervention performed. Exclusion criteria were review studies, articles written in languages other than English and ex vivo studies.

The same reviewers selected the captured studies according to inclusion and exclusion criteria, examining the full text of articles. Any disagreement was assessed by consensus, and a third reviewer (M.F.) was consulted when necessary. The following data were evaluated by two reviewers (F.S. and M.M.) independently: study title, author, year of publication, study design, number of participants, ocular assessments, and outcomes. For unpublished data, no effort was made to contact the corresponding authors. All the selected records were evaluated to assess the strength of evidence according to the Oxford Centre for Evidence-Based Medicine (OCEM) 2011 guidelines and the Scottish Intercollegiate Guideline Network (SIGN) assessment system for individual studies as implemented for Preferred Practice Patterns by the American Academy of Ophthalmology [42,43]. Finally, the quality of evidence-based on the Grading of Recommendations Assessment, Development and Evaluation (GRADE) system was also assessed [44].

From the total of 60 studies identified following the initial literature search, 48 abstracts were identified for screening after the elimination of duplicated records. Forty-two of these met the inclusion/exclusion criteria for full-text review. Eight articles were excluded: six not written in English, 1 review papers and 1 ex vivo studies. After full-text evaluation, 34 studies about bCSCR were included [2,3,4,5,6,7,8,9,10,11,12,13,14,15,17,18,21,22,23,24,25,26,27,28,30,31,32,33,35,36,37,38,39,40].

The included studies were listed in the Appendix A. Ten were retrospective case series, 23 case reports and 1 a prospective observational case series. The following outcomes were assessed about bCSCR: the epidemiology, the pathogenesis and risk factors, the clinical presentation, the imaging features, the differential diagnosis. Finally, the feasibility and main advantages of treatments available were evaluated. The current systematic review reports a qualitative analysis, detailed issue-by-issue in narrative fashion for the heterogeneity of available data and the design of the available studies (i.e., case reports or case series). 

## 5. Conclusions

This rare variant of chronic CSCR is a diagnostic challenge, and an early diagnosis might prevent disease progression from harmful medications as well as unnecessary surgery and visual loss. Treatment of bCSCR is still mattered of controversy, and to date published studies are retrospective and uncontrolled. Failure to differentiate bCSCR from other diseases may result in inappropriate use of corticosteroids, worsening the clinical picture and leading to a poor visual outcome. However, if laser photocoagulation, PDT, TTT or scleral thinning offer any advantage over observation alone is not clear. Further studies are warranted to address this issue, improving the patient’s care.

## Figures and Tables

**Figure 1 pharmaceuticals-13-00221-f001:**
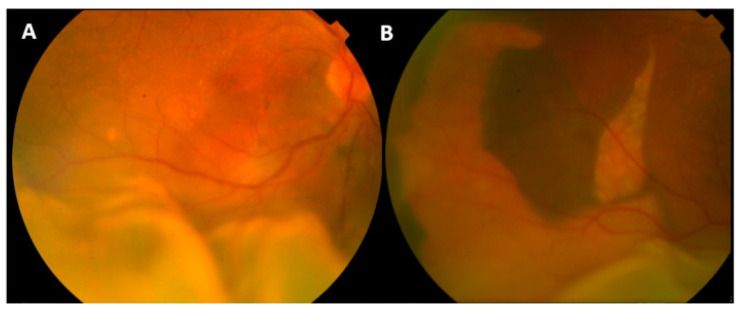
Fundus photographs reveal an exudative inferior retinal detachment macula-off (**A**) with a large retinal pigment epithelial tear in the temporal quadrant (**B**). Reproduced with permission from [14].

**Figure 2 pharmaceuticals-13-00221-f002:**
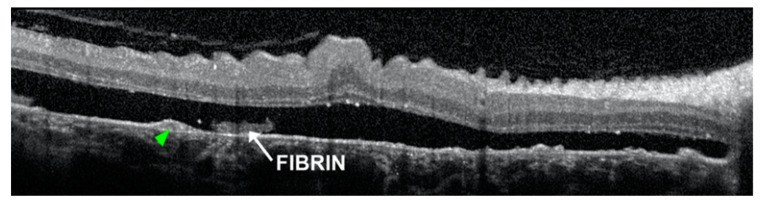
Optical coherence tomography (OCT) detects a pigment epithelial detachment (PED) with internal hyperreflectivity (green arrowheads) beneath the subretinal fluid, containing fibrin. Reproduced with permission from [6].

**Figure 3 pharmaceuticals-13-00221-f003:**
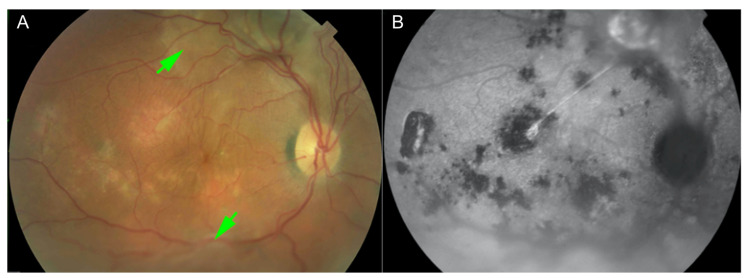
Fundus examination reveals turbid subretinal fluid at sites of neurosensory detachment (green arrows) (**A**); Fundus autofluorescence shows multiple areas of hypo-autofluorescence at the posterior pole (**B**). Reproduced with permission from [6].

**Figure 4 pharmaceuticals-13-00221-f004:**
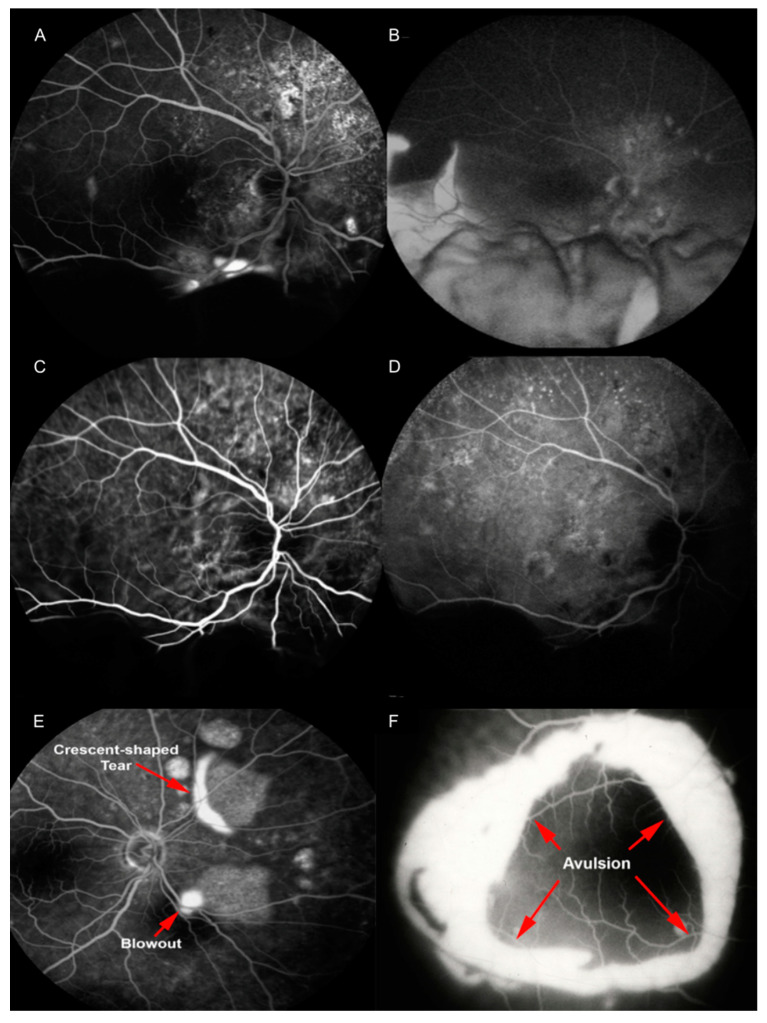
Fluorescein angiography (FA) reveals multiple areas of leakage at the posterior pole in the early phase (**A**); with consequent pooling of the subretinal space and the dependent exudative retinal detachment (**B**). Indocyanine green angiography (ICGA) shows hyperpermeable and dilated choroidal vessels (**C**); in late phase pigment epithelial detachment (PED) it appears still hypofluorescent (**D**). FA shows focal area of hyperfluorescence (blowouts) within PED and an RPE tear with crescent shape (**E**). FA delineates a concentric RPE tear, with a retracted RPE at the center (**F**). Reproduced and merged with permission from [6,14].

**Figure 5 pharmaceuticals-13-00221-f005:**
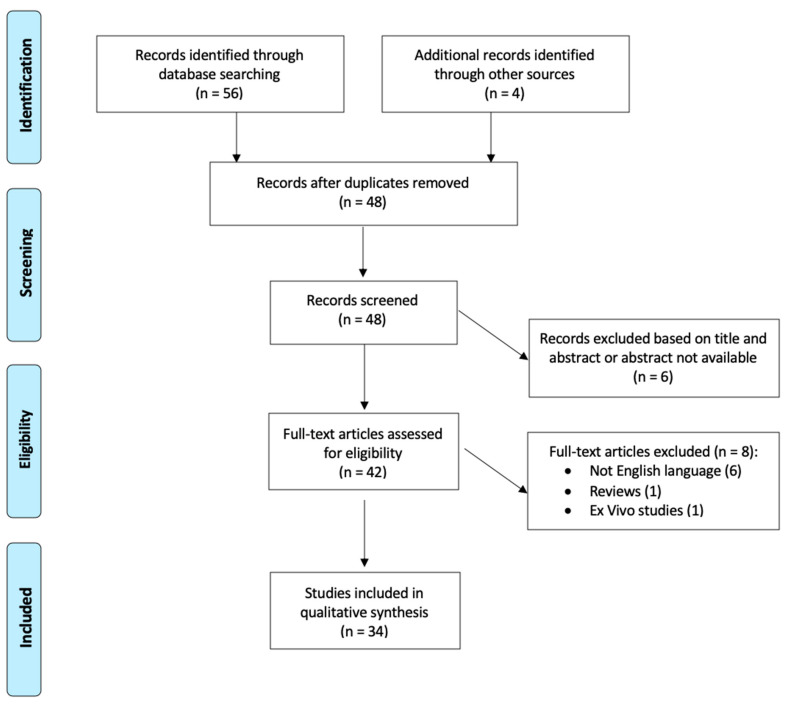
PRISMA flowchart.

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
