# Peer review of "Bullous Central Serous Chorioretinopathy: A Rare and Atypical Form of Central Serous Chorioretinopathy. A Systematic Review"

_pharmaceuticals, 2020, doi:10.3390/ph13090221_

Round 1

Reviewer 1 Report

The manuscript is a comprehensive review which focuses on a clinically important issue. The relevant findings are discussed in a logical order. Some grammatical or spelling errors (lanes 60, 67, 97, 99, 131, 155, 218, 307, 308, 313, 319) should be corrected. Therefore, I recommend this manuscript for publication with minor modifications.

Author Response

We really appreciate the reviewer’s comment. We improved the manuscript as suggested:

Line 60: “studies as implemented for Preferred Practice Patterns by the American Academy of Ophthalmology [5,6].”

Line 67: “After full-text evaluation, 34 studies about bCSCR were included [2,3,8-39].”

Line 97: “It is involved in physiologic transport of ions and macromolecules, and it might cause fluid accumulation into subretinal space [19].”

Line 99: “Besides, induced gluconeogenesis and glycolysis in bCSCR probably promoted cone photoreceptors’ survival [19].”

Line 131: “of thickened neurosensory retina associated with a semi-transparent subretinal space,”

Line 155: “In the phases preceding bCSCR development, fluorescein angiography (FA) reveals focal areas of hyperfluorescence (blowouts) within PEDs [10].”

Line 218: “9 repeated treatment) with resolution within two months. However, the final BCVA was comparable between”

Line 307: “Therefore, prospective randomised controlled trials are warranted in order to evaluate further the efficacy and safety of surgical techniques to treat bCSCR.”

Line 308: “Anti-VEGF intravitreal injections” (VEGF has been spelt previously in the revised manuscript).

Line 313: “Besides, peripheral neovascularization complicating bCSCR have been successfully treated with argon laser photocoagulation of leakage sites [23].”

Line 319: “although bCSCR seems to be the result of environmental, genetic, and biomedical interactions.”

Reviewer 2 Report

This review article discussed the bullous central serous chorioretinopathy (bCSCR), a rare variant of the central serous chorioretinopathy (CSCR). In this systemic review, pathophysiology, clinic presentation and treatment methods of bCSCR were discussed.

1, Please discuss more the difference between bCSCR and typical CSCR, such as possible pathophysiology, clinic presentation, and treatment strategies.

  1. As optical coherence tomography angiography (OCTA) is widely used in the diagnosis of CSCR with some specific characters on this disease, please discuss more this topic.

Author Response

This review article discussed the bullous central serous chorioretinopathy (bCSCR), a rare variant of the central serous chorioretinopathy (CSCR). In this systemic review, pathophysiology, clinic presentation and treatment methods of bCSCR were discussed.

  1. Please discuss more the difference between bCSCR and typical CSCR, such as possible pathophysiology, clinic presentation, and treatment strategies.

We thank the reviewer for the comment and to give the opportunity to discuss the differences between CSCR and bCSCR. We modified the pathophysiology section as follow: “Choroidal vascular dysfunction is a crucial feature in the bCSCR pathophysiology, as in CSCR [1]. In particular, marked congestion and breakdown in the choroidal vessels' permeability alter RPE, damaging RPE outer blood-retina barrier [1]. Subsequently, proteins and fibrinogen leaks, thus a PED and/or a subretinal exudation occur [1]. Both chronic CSCR and bCSCR shows pachychoroid feature as outer choroidal vessel dilation, inner choroid atrophy, and choroidal hyperpermeability; however, to date, the mechanism responsible for bCSCR development is still unclear [1,18]. It has been speculated that sub-EPR fibrin and increased hydrostatic pressure internal to PED provoke circumferential traction on the RPE layer and a subsequent RPE tear [10].

Consequently, the intense subretinal fluid accumulation, rich of fibrin and weighty, provokes an ERD, whose fluid shifts according to the patient's position [8]. Besides, steroid (systemic or epidural) may play a role in impending the healing of tear and increasing choriocapillaris’ permeability [17]. Nevertheless, no difference in corticosteroid exposure was reported between bCSCR and chronic CSCR patients [10].“

Besides we added a paragraph at the end of clinical presentation: “Comparing chronic CSCR and bCSCR, best-corrected visual acuity at the baseline and the final visit shows no statistical difference in a retrospective review of patients [10]. Although RPE tear and peripheral nonperfusion occur only bCSCR and also PEDs are significantly more frequent [10]. Besides, on cursory examination, the ERD can simulate diseases not belonging to the CSCR spectrum, leading to inappropriate therapeutic procedures [18].”

Finally, we added a paragraph at the beginning of treatment section to compare chronic CSCR and bCSCR therapeutic approach: “As in CSCR, the treatment aims to preserve the outer neurosensory retinal layers, achieving a lasting and complete SRF resolution, because even a small amount of SRF can damage the photoreceptors [43]. Elimination of risk factors is advisable in bCSCR as in CSCR. In particular, systemic corticosteroid withdrawal is suggested as the first step, if possible, according to systemic conditions (e.g. previous organ transplant) [13,21,26,29-32]. Kunavisarut et al. stopped steroid medication in six patients and tapered it in one (adding azathioprine). Retinal reattachment occurred in 5 months (range 1-9 months) in 86% of cases [30].

The treatment of chronic CSCR is still debated, but recently evidence-based guidelines have been proposed [43]. After individuation and elimination of risk factors, if persistent SRF and one or more focal leakage points on FA are present, ICGA- and FA-guided half-dose (or half fluence) verteporfin photodynamic therapy (PDT) is suggested as first-line treatment. Then if SRF persists FA, ICGA and OCTA should be obtained. If choroidal neovascularization is detected, intravitreal injections should be performed, with or without adjunctive PDT. If leakage (diffuse or focal) is noted on FA/ICGA, the following treatment can be performed: retreatment with PDT, subthreshold micropulse laser, mineralocorticoid receptors antagonist, observation. Finally, if ICGA hyperfluorescence or leakage on FA are not detectable, retreatment is not advisable, because no concrete evidence is available [43].

Conversely, the bCSCR therapeutic algorithm is still a matter of controversy, due to its rare frequency and the lack of randomized clinical trial to compare different treatment results. After risk factors removal, if laser photocoagulation on focal leakage seen on FA can be performed [12,24,32]. Nevertheless, if the leakage area is extended, laser photocoagulation can lead to scotomata. Therefore other therapeutic approaches have been described, such as PDT, oral eplerenone, and transpupillary thermal therapy (TTT) [33]. Finally, anti-vascular endothelial growth factor (VEGF) intravitreal injections and surgery have been reported in the literature [18,22].”

  1. As optical coherence tomography angiography (OCTA) is widely used in the diagnosis of CSCR with some specific characters on this disease, please discuss more this topic.

We understood the reviewer’s point, so we improved the OCTA paragraph as follow: “Finally, OCT angiography (OCTA) can detect choroidal neovascularization in chronic CSCR; however, its role in bCSCR is still being explored [1]. In particular, OCTA shows ill-defined low detectable flow areas (dark areas) corresponding to SRF and well-delineated areas with no detectable flow (dark spots) at the choriocapillaris level, corresponding to PED [1]. Besides, OCTA depicts the absence of any vascular network below the RPE tear [18]. Finally, OCTA highlights the pachychoroid pattern in bCSCR, as well-delineated, high-flow, tangled pattern areas within the choriocapillaris overlying dilated outer choroidal vessels [18].”

Reviewer 3 Report

Excellent review, well-written about a very difficult topic: Bullous central serous chorioretinopathy. Congratulations to the authors. 

Please correct/clarify the following points:

Line 76: (84% respectively 7.4%). Please clarify this sentence.

Line 233: improved from 20/60 to 20/80. Please correct the mistake.

Author Response

Excellent review, well-written about a very difficult topic: Bullous central serous chorioretinopathy. Congratulations to the authors.

We really appreciated the reviewer’s comment; we improved the manuscript as suggested.

Please correct/clarify the following points:

Line 76: (84% respectively 7.4%). Please clarify this sentence.

We understood the reviewer’s point, so we modified the sentence accordingly: “Bilateral involvement is more frequent in bCSCR compared to chronic CSCR (84% respectively 7.4%)  [8,9].”

Line 233: improved from 20/60 to 20/80. Please correct the mistake.

We are sorry for the mistake, we double-checked the reference, so we modified the sentence accordingly: “A complete resolution of the retinal detachment was obtained, although BCVA decreased from 20/60 to 20/80 seven months after the treatment, due to the disruption of the photoreceptor inner segment/outer segment reflective band within the fovea. A pigmented retinal scar was also developed at the site of the RPE tear [22].”
